# Protection of Mono and Polyunsaturated Fatty Acids from Grapeseed Oil by Spray Drying Using Green Biopolymers as Wall Material

**DOI:** 10.3390/foods11243954

**Published:** 2022-12-07

**Authors:** Diego Mauricio Sánchez-Osorno, Angie Vanesa Caicedo Paz, María Camila López-Jaramillo, Aída Luz Villa, Julián Paul Martínez-Galán

**Affiliations:** 1Grupo de Investigación Alimentación y Nutrición Humana-GIANH, Escuela de Nutrición y Dietética, Universidad de Antioquia, Cl. 67, No 53-108, Medellín 050010, Colombia; 2Grupo de Investigación e Innovación Ambiental GIIAM, Institución Universitaria Pascual Bravo, Cl. 73, No 73a-226, Medellín 050034, Colombia; 3Grupo Catálisis Ambiental, Universidad de Antioquia, Cl. 67, No 53-108, Medellín 050010, Colombia

**Keywords:** fatty acids, spray drying, grapeseed oil, encapsulation, microcrystalline cellulose, whey protein isolated

## Abstract

One of the most common ways to protect oils is microencapsulation, which includes the use of encapsulating agents. Due to the environmental problems facing humanity, this study seeks to combine green biopolymers (microcrystalline cellulose and whey protein isolate) that function as encapsulating agents for grapeseed oil. Grapeseed oil that is obtained from agro-industrial waste has shown health benefits, including cardioprotective, anticancer, antimicrobial, and anti-inflammatory properties. These health benefits have been mainly associated with monounsaturated (MUFA) and polyunsaturated (PUFA) fatty acids. In this sense, it has been observed that grapeseed oil can be easily modified by environmental factors such as oxygen, high temperatures, and light, showing the instability and easy degradation of grapeseed oil. In this study, grapeseed oil was encapsulated using the spray-drying technique to conserve its lipidic profile. Powder recovery of the grapeseed oil microcapsules ranged from 65% to 70%. The encapsulation efficiency of the microcapsules varied between 80% and 85%. The FTIR analysis showed chemical interactions that demonstrate chemisorption between the grapeseed oil and the encapsulating material, while the SEM micrographs showed a correct encapsulation in a spherical shape. Gas chromatography showed that the lipid profile of grapeseed oil is preserved thanks to microencapsulation. Release tests showed 80% desorption within the first three hours at pH 5.8. Overall, whey protein and microcrystalline cellulose could be used as a wall material to protect grapeseed oil with the potential application of controlled delivery of fatty acids microcapsules.

## 1. Introduction

There is now a growing trend towards the use of healthy foods to preserve and improve health, or to prevent various types of diseases. This consumer trend has been reflected in the food industry, which seeks to respond to these market needs by offering functional and nutraceutical foods. One of the most notorious changes in healthy eating has been the substitution of fats of animal origin for vegetable oils, since the latter have healthier characteristics. In this sense, grapeseed oil exhibits a high concentration of polyphenols, which are antioxidant substances that function as preservatives in food or as protective agents at the cellular level against oxidative damage from free radicals [1].

Grapeseed oil has a high content of monounsaturated fatty acids (MUFA) and polyunsaturated fatty acids (PUFA) [2]. These fatty acids have been associated with health benefits, including antimicrobial, cardiac protection, anti-inflammatory, and anticancer properties [3]. It is possible to improve the nutritional value of some food products by incorporating grapeseed oil in various matrices, thus taking advantage of the beneficial effects it has on health. However, some industrial processes such as the use of high temperatures, other compounds present in food, light, and oxygen, could generate a degradation or modification of grapeseed oil because it is chemically unstable and susceptible to oxidative degradation. These instabilities are shown as unwanted flavors, colors, aromas, and the loss of its nutritional value [4].

Microencapsulation is a technology that seeks to protect bioactive compounds. In the case of lipids, it can delay oxidation, reduce volatility, and improve the stability of oils and aromas. Microencapsulation consists of coating a particle that is in a solid, liquid, or gaseous state. This coating consists of a thin layer that isolates the particle from external factors and keeps it completely contained within the wall of the capsule. The microencapsulation principle protects a bioactive compound from external factors that can facilitate degradation, creating a physical barrier between the active compounds and the outside [5]. Although there are many methods that allow the encapsulation of bioactive compounds, spray drying is the most widely used, in industries such as food, pharmaceuticals, and cosmetics, because it is an inexpensive, flexible, and easily scalable process. This technique allows a liquid mixture, in which the bioactive compound and the encapsulating agent are found, to become fine droplets that will be dehydrated with the help of a stream of hot air to obtain microparticles in the form of powders. By having the bioactive compounds in powder form, it is possible to improve storage, transportation, and their dosage in food formulations [6].

There is a wide variety of encapsulating agents. Among the most used encapsulating agents for lipids are Arabic gum, barley protein, carrageenan, chitosan, maltodextrin, methylcellulose, skim milk powder, and whey protein [7]. Those are biodegradable, abundant, and generally non-toxic. On the other side, it is possible to find synthetic polymers used as encapsulating agents, as well as aliphatic polyesters, such as poly (lactic acid) and copolymers of lactic and glycolic acids [7]. The disadvantage of these materials is that synthetic polymers show a lack of biocompatibility, and they have a high potential for contamination. However, there is a growing inclination for environmentally friendly encapsulating agents during their extraction, production, and final disposal processes. These environmentally friendly encapsulating agents are called green biopolymers. Green biopolymers are polymers obtained from products or processes that reduce or eliminate the use or generation of substances that are dangerous to humans, plants, or the environment [8,9].

In this sense, obtaining encapsulating agents from agro-industrial waste becomes a great alternative to contribute to the environment. Whey protein concentrate is an encapsulating material extracted from dairy industry waste with good emulsifying properties, high solubility, and low viscosity. This material has a wide range of solubility from pH 2 to pH 9. However, the pH of the solution tends to affect the nature and distribution of the net charge of proteins. Normally, proteins are more soluble at low pH (acids) or high pH (alkalines) due to the excess of charges of the same sign, which produces repulsion between the molecules, thus contributing to a high solubility [10]. Due to its wide range of solubility, whey protein can be considered an immediate release encapsulating material. As for cellulose, it is extracted from agro-industrial waste such as wood, banana rachis, and oil palm rachis, among others [11]. Therefore, recent investigations have focused on the development of cellulose micro-particles, known as microcrystalline cellulose (MCC), with prominent characteristics, making it useful for a wide range of materials and products [12,13].

MCC is a partially depolymerized, pristine cellulose derivative conventionally prepared through an acid hydrolysis approach to eliminate the disordered regions, whereas the ordered domains that present a higher resistance to the hydrolysis process remain intact [14]. The obtained MCC shows short particles and exhibits a high crystallinity index, improved density, a large surface area, and increased thermal stability [14]. Besides that, the presence of several accessible and active hydroxyl side chemical groups on the surface allows MCC to be functionalized through multiple energetic chemical groups, such as nitrate esters, nitramines, tetrazoles, and azides, to design innovative high-value biopolymers for advanced energetic materials [15] with the ability to interact with other encapsulating agents thanks to its availability of OH groups and its hydrophilicity. The importance of microcrystalline cellulose (MCC) in the encapsulation process lies in its use as a viscosity modifier and as a thickener [11]. Additionally, MCC is compatible with a large number of food compounds, including carbohydrates and hydrocolloids, due to its ability to retain flavor and its solubility in water [16].

To preserve the lipid profile of grapeseed oil over time, this work aimed to use green biopolymers to protect the monounsaturated and polyunsaturated fatty acids present in grapeseed oil, and thus preserve its nutritional characteristics. For this purpose, a whey protein (WPI) to microcrystalline cellulose weight ratio of 3:1 was evaluated to protect the grapeseed oil by using spray drying as the selected encapsulation method.

## 2. Materials and Methods

### 2.1. Materials

#### 2.1.1. Isolation of MCC

For MCC, the methodology developed in the Rojas patent was followed with some modifications [17]: 280 g of an agro-industrial waste (cane bagasse, fique, rice husks, and corn cobs) and HCl solution (1 N). A 5 L flask was used (60 min room temperature), and the mixture was brought to a boil for another 1.5–2 h under constant stirring. The mixture was cooled and filtered. The white residue obtained was washed and then air-dried (yield > 85%). A mixture of an aqueous solution of sodium hydroxide and dry powder was prepared with a constant application of stirring. The previous mixture was left to stand between 4 and 12 h at room temperature. Ethanol was added until a white precipitate was obtained, which normally occurs when a concentration between 50 and 60% of ethanol is reached. The solid obtained was air-dried and passed through a US#20 sieve. The sieved material was dried to a moisture of less than 6% using temperatures between 45 and 50 °C (yield > 90%) [17,18].

#### 2.1.2. Whey Protein Isolate

WPI with 80% protein content, on a dry basis, was supplied by Bell Chem International S.A.S. (Rionegro, Colombia). Grapeseed oil was supplied by Distriol (Sao Paulo, Brazil). Absolute ethanol of analytical grade was from Merck (Rahway, NJ, USA). Deionized water (Milli-Q) was used for all preparation.

#### 2.1.3. Preparation of Emulsions

WPI and MCC were dissolved in water under stirring using an Ultra-Turrax T-25 homogenizer (IKA model Bioblock Scientific, Medellín, Colombia) for 5 min at room temperature, and grapeseed oil was then added. The emulsion was prepared by adding the lipid component to the aqueous phase using an Ultra-Turrax operated at 25,000 rpm for 10 min and then passed to spray drying. The weight ratio of encapsulating agent to grapeseed oil was 3:1, according to other investigations [1,19,20], and confirmed in laboratory tests.

### 2.2. Encapsulation Using Spray-Drying

Immediately after preparation of the feed solution, encapsulation was carried out using a Mini Spray Dryer (Buchi B-290, Pilotech, Shaghai, China). The process conditions were as follows: a nozzle diameter of 0.5 mm, a drying chamber diameter of 35 cm, and a height of 90 cm. Parameters included the pump (10%), aspirator (100%), flow rate (600 L/h), inlet temperature (180 °C), and outlet temperature (100 °C).

Each experiment was performed by preparing 300 mL of emulsion. Drying was performed in triplicate.

The microspheres were evaluated for solubility, porosity, encapsulation efficiency, particle size, and morphology. The protective effect of spray-drying was verified by a compositional analysis of grapeseed oil, before and after encapsulation, via gas chromatography.

### 2.3. Surface Free Oil Content and Microencapsulation Efficiency

The amount of unencapsulated oil (i.e., free oil or surface oil) present at the surface of the powders was measured using a method described by Tan [21], with some modifications. For this purpose, 2 g of microparticles were mixed with 15 mL of hexane, placed in a glass vial with a lid, and stirred at room temperature for 2 min using a vortex to extract the free oil. This was followed by a decantation and filtration process. Subsequently, the washed microparticles were rinsed with hexane 3 times using 20 mL of hexane. The residual solvent was removed by drying at 60 °C until brought to a constant temperature. The weight difference in the powder before and after hexane extraction was used to calculate the free oil content as a percentage. Microencapsulation efficiency (EE) was calculated using Equation (1).
(1)$$Microencapsulation\;efficiency=\left[\frac{\left(total\;oil-surface\;oil\right)}{total\;oil}\right]\times 100$$

### 2.4. Microcapsule Characterization

#### 2.4.1. Solubility

The solubility of spray-dried microparticles was measured according to Jafari [22], with some modifications. The standard technique for dairy powders was used, initially preparing a 5% (*w*/*w*) solution in distilled water. To obtain a good dispersion, stirring was used at 750 rpm for 30 min at 20 °C. An aliquot of 40 mL was taken and centrifuged for 10 min at 1000 rpm by using an CD-0412-50 centrifuge. Five grams of the supernatant were dried at 105 ± 5 °C for 24 h, transferred to a desiccator, and reweighed. The amount of soluble material in the supernatant was calculated and expressed as the solubility percentage [22].

#### 2.4.2. Porosity

Powder porosity was measured according to Jafari [22] to determine the bulk density. Two grams of microcapsules were placed in a cylinder (10 mL), and the cylinder was tapped several times on a rubber surface. The apparent density was determined with the mass-occupied volume relationship. The true density was then calculated by filling approximately 1 g of spray-dried whey powders in a burette containing toluene [22]. The rise in toluene level (mL) was measured, and true density was calculated according to Equation (2):(2)$$True\;density=\frac{weight\;of\;powder\;sample\;\left(g\right)}{rise\;in\;toluene\;volume\;\left(mL\right)}$$

Finally, the porosity of the powder samples was calculated using the relationship between the bulk and the true density of the powders, as shown in Equation (3) [23]:(3)$$Porosity=\left(1-\frac{\left(bulk\;density\right)}{\left(true\;density\right)}\right)$$

#### 2.4.3. Fourier Transform Infrared Spectroscopy (FTIR)

FTIR measurements were performed in triplicate using an Alpha Platinum FTIR-ATR spectrometer (range: 4000–600 cm^−1^; number of scans: 144, zero filling factor: 4, resolution: 4 cm^−1^, mode: absorbance). The powder was settled on the optic window with a diamond crystal.

#### 2.4.4. Morphology SEM and TEM

The morphology of the spray-dried microcapsules was observed with a scanning electron microscope (SEM, JSM-590LV, JEOL), operating at 15 kV. The microcapsules were coated with gold and observed under the microscope. Additionally, the internal morphology of the microparticles was observed by initially freezing the microparticles with nitrogen and then fracturing them. The interior morphology of the wet status microcapsules was also observed using a transmission electron microscope (TEM, JEM 1011 JEOL) at an accelerating voltage of 120 kV. The microparticle samples were diluted in hexane (1:100 *v*/*v*), and the suspension was kept in an ultrasonic bath (Unique, model USC-1800, Colombia) for 30 min. Subsequently, the suspension was deposited on a copper grid coated with carbon film. The images were observed at a magnification of 120 k.

#### 2.4.5. Particle Size Distribution

SEM micrographs (1000× magnification) were analyzed using ImageJ software. For this analysis, at least 140 particles were taken to measure their diameter. From these measurements, a global average was performed, and the size distribution curves were constructed.

### 2.5. Gas Chromatography

Analysis of the composition of grapeseed oil, before and after encapsulation, was performed via gas chromatography-mass spectrometry (GC-MS: Varian CP 3800 and CP-Sil 8 CB Low Bleed/MS column (30 m × 0.25 mm × 0.5 µm)). Equipment conditions were as follows: injector: 250 °C; helium: 1.5 mL/min; oven: from 50 °C to 240 °C at 3 °C/min. The composition of the oil was determined by mass spectrometry (ion trap, temperature at 220 °C; temperature other than 80 °C, transfer line temperature at 240 °C).

Gas chromatography (GC) was used to perform the analysis of FAME (Fatty Acid Methyl Ester) in hexane (Agilent Technologies 6890, Hewlett-Packard, Avondale, PA, USA) equipped with a split/splitless injector. The flow rate was 128 mL/min, and the temperature was 150 °C (5 min) to 220 °C (30 min) at a rate of 4 °C/min. The temperature of the injection was 250 °C.

#### 2.5.1. Fatty Acid Profile of Grapeseed Oil

The FAMEs (Fatty Acid Methyl Esters) were analyzed by gas chromatography according to the American Oil Chemists’ Society Method (AOAC, 2005). The analysis of fatty acid profile was carried out by gas chromatography (GC). FA methyl esters were analyzed on an Agilent 6890N gas chromatograph with a flame ionization detector (FID), a TR-CN100 capillary column, a 60 m × 250 µm × 0. 20 µm ID, a split/split less injector with a split ratio 100:1, an injection volume of 1.0 uL, an injector temperature of 260 °C, an oven temperature program from 90 °C (7 min) to 240 °C (15 min) at a rate of 5 °C/min, and a detector temperature of 300 °C. The carrier gas saw a flow rate of 1.1 mL/min. A standard (37-component FAME Mix, Supelco) was used for fatty acid identification. The results are presented as the relative amount of each fatty acid (% of FA/100 g of sample). The concentrations of total SFA, MUFA, and PUFA were calculated as the sum of the corresponding families. The percentage of total fat was determined as the sum of SFA, MUFA, and PUFA.

#### 2.5.2. Fatty Acid Profile of Microencapsulated Grapeseed Oil

The microcapsules of grapeseed oil were used as a solid matrix. Therefore, prior to the determination of the fatty acid profile, acid hydrolysis on the oil microcapsules was carried out using a SOXCAP 2047. Fat extraction was then performed using the SOXTEC 2050 (FOSS, 2017). After extraction, hexane was added with a pasteur pipette, and the contents were transferred to an Erlenmeyer flask, where the methylation process continued. This procedure was repeated three times. The methylation process was followed as described above.

### 2.6. Controlled Release

To determine the release profile of the microcapsules, the dynamic dialysis method was used. Grapeseed oil capsules (3 mg mL^−1^) were scattered in 100 mM PBS (phosphate buffer solution) at pH 1 (stomach) and pH 5.8 (intestine). This experiment was performed using dialysis bags, at 37 °C, for incubation in absolute ethanol (30 mL), and this was stirred at 100 rpm. One-milliiter aliquots were withdrawn at various time intervals, and the same volume was replaced with absolute ethanol. The visible ultraviolet spectroscopy technique was used to determine the released oil % [24,25]. The concentrations of the grapeseed oil were determined using the standard graph (y = 0.0339x + 0.0332, R^2^ = 0.9921). Samples were analyzed in triplicate. The cumulative release rate was obtained by the ratio of cumulative release amount to the amount of grapeseed oil encapsulated.

### 2.7. Statistical Analysis

The data collected was reported as the average of tests performed in triplicate. ANOVA was used for the analysis of results (SPSS program version 10.0 for Windows). Duncan’s multiple range test was used to test for differences between means.

## 3. Results

### 3.1. Surface Free Oil Content and Microencapsulation Efficiency (EE)

Spray drying was used to produce microparticles using two green polymers, MCC and WPI. EE% was found in 86.9% of the grapeseed oil microencapsulated, which is higher than those results obtained in the encapsulation of grapeseed oil with Arabic gum (67.92%) and Arabic gum plus maltodextrin (63.47%) [1]. Encapsulation efficiencies between 87% and 92.1% have been associated with droplets with diameters between 119.8 and 217.6 nm [26]. Additionally, the good encapsulation efficiency obtained using MCC and WPI showed a correct viscosity in the emulsion, since the increase in oil retention was achieved when there was a rapid formation of the drop and short exposure times in the formation of the surrounding crust of the atomized drop during the drying process [26]. Encapsulation by spray drying using WPI showed encapsulation efficiency percentages above 98% [27]. The decrease in WPI encapsulation efficiency in this study could be due to mixing with MCC.

### 3.2. Characterization of the Grapeseed Oil Microcapsules

Solubility in powders is considered the determining key for its reconstitution in the dissolution process. The solubility of WPI has been reported between 79.29% and 82.24%, while the solubility of produced microcapsules containing grapeseed oil was 65% ± 0.26 [27]. This result showed that the addition of MCC to WPI decreases both the encapsulation efficiency and the solubility of more microcapsules. The effect of MCC and WPI could be because the low solubility of microcrystalline cellulose in the wall material reduces the solubility of the microcapsules, since the cellulose–protein and protein–protein interactions increase because the electrostatic forces are at a minimum, and less water interacts with the protein molecules [28,29]. Additionally, an influence of the temperature used in spray drying on solubility has been demonstrated, especially with WPI. Inlet temperatures used in the spray dryer between 180 and 190 °C has shown a decrease in the solubility of whey protein [22]. This investigation was carried out using the previously reported temperatures. Finally, the low solubility can be related to the formation of a surface layer over the powder particles, thereby reducing the diffusion rate of water molecules through the surface of particles [22].

Porosity is an important factor in determining the functionalities of agglomerates. Porous microparticles are widely desired as suitable materials for carriers for the drug delivery system. Porous microspheres’ low density and large surface areas are inherent properties of great importance [30]. These unique properties are especially important for gastrointestinal delivery systems. Since fatty acids are mainly absorbed in the small intestine, obtaining porous microspheres is widely desired. In this research, the porosity of the powder sample was 0.45% ± 0.28. Both WPI and MCC obey mesoporous materials with pore sizes between 2 and 50 nm, enabling the creation of porous particles [30,31].

The high porosity value could be due to formation of more vacuoles inside powder particles at higher temperatures, which results in a lower bulk density and a higher porosity [32]. An increase in porosity related to the addition of WPI was observed during the spray-drying process. This increase in porosity may be related to the increase in particle size, which increases with increasing WPI.

FT-IR analysis was applied to assess the properties of the microcapsules of grapeseed oil. The spectra of the grapeseed oil, WPI, MCC, and the microcapsules are presented in Figure 1.

The MCC spectrum shows the representative peaks in which the region from 3400 to 3500 cm^−1^ and the absorption at 2900 cm^−1^ is due to stretching of –OH groups and CH_2_ groups, respectively [33]. The absorption band at 1163 cm^−1^ and the peak at 896 cm^−1^ correspond to C–O–C stretching and are associated with the C–H rock vibration of cellulose (anomeric vibration, specific for β-glucosides) observed in MCC [34]. Additional bands are shown in Table 1.

The whey protein spectrum revealed fat and oil regions between 3000 and 2800 cm^−1^, which corresponds to the C-H stretching vibrations of carbonyl groups of triglycerides [35]. Other peaks related with fatty acids are located at 2924 and 2854 cm^−1^ [36]. The main vibrational bands analyzed in the region between 1700 and 1200 cm^−1^ were fatty acids, polysaccharides, and proteins. Between 1700 and 1500 cm^−1^, it is possible to find the Amide I (ν C=O, ν CeN) in 1640 cm^−1^ and Amide II in 1550 cm^−1^ (δ NeH, ν CeN), related to peptide bonds (CO-NH) [37]. The region dominated by polysaccharide peaks is located at 1200–900 cm^−1^ [38]. Another important band is shown in Table 1.

In the microencapsulated grapeseed oil, deconvolution was used to find the main bands of the different encapsulant agent and the oil to find the formation of new peaks. The grapeseed oil spectrum showed the adsorption bands for common triglycerides and fatty acids, and the main peaks are shown in Table 1.

Main bands corresponding to MCC, WPI, and grapeseed oil were observed in the microcapsule spectra, and no new peaks were found. In the microcapsule spectrum, two regions that are related to fatty acids, especially linoleic and oleic fatty acids, received special attention. The first lipid region is located between 3050 and 2800 cm^−1^, and the second lipid region is located between 1800 and 1350 cm^−1^. In these regions, it is possible to find the bands mainly associated with lipids. The band at 3006 cm^−1^ results from the C-H stretching vibration of HC=CH groups, which could be a useful indicator of the different degrees of unsaturation in acyl chains of phospholipids [39,40]. The absorption band of carbonyl stretches of carboxyl at 1747 cm^−1^ exhibited the presence of lipids [41], assigned to the C=O stretching vibration of ester groups in triacylglycerols [42].

These results are in agreement with other studies that reported the IR regions of edible oil for unsaturated fatty acids featured around 2900 cm^−1^, as well as oleic acid and linoleic acid profiles [43].

FTIR analysis showed no new peak formation in the mixture, which is evidence of physical adsorption. Additionally, the presence of the characteristic peaks corresponding to the encapsulated fatty acids show that there is a protection provided by the encapsulating materials [44]. Finally, the physical adsorption observed between the different materials is provided by secondary interactions such as hydrogen bonds and Van der Waals forces [45]. The foregoing allows us to suppose that the opening of the microcapsules to release the encapsulated grapeseed oil can occur under the conditions of the gastrointestinal system, where factors such as pH, temperature, and agitation are present.

Figure 2 shows the different micrographs (SEM) of the microcapsules. All the samples observed show a superficial shrinkage of the particles, similar to the illustrations of other authors [46]. It has been reported that the use of optimal encapsulation conditions produces particles with smooth surfaces. However, when changing the encapsulating materials, it is necessary to modify the encapsulation conditions. The cellulose fiber used in encapsulation processes produces rough surfaces [47], which would explain the roughness present in the obtained particles.

Additionally, we believe that there is an impact of temperature on surface roughness and particle shrinkage as reported by Yang (2021), where three states associated with temperature are identified. At the beginning of the drying process, in the water drop formation, there is a greater evaporation of water from the drop compared to the wall material. At this point, it can be determined that the pressure inside the drop increases, which generates an expansion in volume. In the next stage, the drying and subsequent hardening of the wall material occurs, producing a cavity inside the particle. In this way, we have a particle with low permeability but subjected to high temperatures, thus allowing the internal pressure of the particle to be maintained. Finally, in the last stage of drying, the temperature decreases, causing a decrease in the internal pressure of the particle, in turn causing the surface of the particle to contract, and the morphology observed in the microparticles of this research is thus obtained [46].

The microcapsules do not show cracks or fractures, which is important for the protection and retention of the encapsulated oil. However, when evaluating the internal morphology (Figure 2D), a correct wall formation is observed. Additionally, in the hollow part there is evidence of droplets embedded in the matrix, this being typical of the spray-drying process [48]. The wall thickness obtained is consistent at the drying temperatures used, exhibiting a good thickness in the wall material [48]. Although the exterior of the particles has a smooth appearance, porosities are observed in the formation of the wall, which is consistent with the results obtained in the porosity analyses. This confirms that the microcapsules formed by WPI and MCC are effective for the encapsulation of grapeseed oil.

The understanding of the internal structure of the microparticle as well as the dispersion of the particles within the matrix of the wall material was analyzed by transmission electron microscopy (TEM). Qualitative analysis of the microparticles was performed based on TEM micrographs (Figure 3). The distribution of grapeseed oil within the coating material, i.e., the green polymers (WPI and MCC), can be seen in Figure 3. As can be seen, the grapeseed oil (light gray color) was concentrated in the center of the particle, being surrounded by the wall material (dark gray color) [49]. These results are according with SEM results where the formed wall has a good formation.

Particle size distribution of microcapsules is shown in Figure 4. A heterogeneous distribution can be observed with a peak that represents a predominant size, but this only indicated the presence of micrometric particles and exhibited monomodal behavior.

### 3.3. Gas Chromatography

Fatty acid profiles (Figure 5) of the grapeseed oil are shown in Table 2. The results show similar ranges to other reports [50]. The fatty acid compositions of the non-encapsulated and encapsulated oil were 16.5% and 21.6% saturated fatty acids (SFAs), 25.67% and 25.47% monounsaturated fatty acids (MUFAs), and 57.81% and 52.89% polyunsaturated fatty acids (PUFAs).

For both, non-encapsulated and encapsulated grapeseed oil, among the observed SFAs, the predominant fatty acid was palmitic acid (C16:0) followed by stearic acid. Low concentrations of other SFAs were found as follows: C14:0: myristic acid; C17:0: margaric acid; C20:0: arachidic acid; C22:0: behenic acid.

Oleic acid (C18:1) was the primary MUFA with 25.37% and 25.08% for non-encapsulated and encapsulated grapeseed oil, respectively. When performing PUFA analysis, it was found that linoleic acid (C18:2) was the main fatty acid, accounting for 51.64% and 47.30% of total PUFAs for non-encapsulated and encapsulated grapeseed oil, respectively. These results are in agreement with previous studies on the fatty acid compositions of other grape seeds [50,51,52].

MUFAs and PUFAs are the predominant fatty acids observed in the fatty acid profile of grape seeds (Table 2). However, the minimum value of the PUFA/SFA ratio recommended is 0.45 [53], which is lower than those found in this study. The highest PUFA/SFA ratio was obtained from non-encapsulated grapeseed oil (3.50), whereas the lowest values were found for encapsulated grapeseed oil (2.44). MUFAs have been studied in past decades due to the beneficial effects on cardiovascular heart disease [54]. Low-density lipoproteins cause the accumulation of plaque in the arteries or arteriosclerosis, and this problem can be reduced with a diet rich in MUFAs, and grapeseed oil has a high content.

### 3.4. Controlled Release

To improve uniformity in the release of the microencapsulated grapeseed oil, and to reduce particle aggregation, the buffer used was adjusted to 24% ethanol. Figure 6 indicates the release of the grapeseed oil-loaded microcapsules at pH 1 and 5.8. When comparing the release obtained at both pHs, a better release was observed at pH 5.8. It was observed that, during the first three hours and at a pH of 5.8, the microparticles of WPI and MCC released 80% of the encapsulated grapeseed oil. At pH 1, the release was around 20% for grapeseed oil. This might be attributed to the interaction among cellulose fibers, which tend to bond with each other at acidic pH [55]. The previous results suggest that the encapsulation system (WPI and MCC) is sensitive to pH, which has applications in pharmacology and functional foods. In this case, the grapeseed oil released from the microcapsules (WPI-MCC) can be released in greater quantity in the small intestine or in the blood system (at pH 5.8–7.4) compared to the grapeseed oil that can be released in the stomach (at pH 1–2). It is conceivable that much of the grapeseed oil will be absorbed into the blood system more efficiently than when the oil is not encapsulated in the microparticles.

Controlled release results cohere with the FTIR results, where we found physic interactions among wall materials and weak interactions with grapeseed oil. Additionally, the low release at pH 1 is consistent with the cellulose reaction at low pH, where cellulose tends to bind. Cellulose at high pH tends to open the net [56], which is consistent with the solubility results where the solubility of WPI and MCC is lower than the solubility of WPI, which explains the controlled release of grapeseed oil during the desorption tests.

## 4. Conclusions

Matrices made by using whey protein and microcrystalline cellulose offer important possibilities for the microencapsulation of grapeseed oil, with higher encapsulation efficiency of 86.9%. EE% was found in 86.9% of the grapeseed oil microencapsulated which is higher than those results obtained in the encapsulation of grapeseed oil with Arabic gum (67.92%) and Arabic gum plus maltodextrin (63.47%). The solubility of produced microcapsules containing grapeseed oil was 65% ± 0.26. The solubility results showed that the addition of MCC to WPI decreases both the efficiency and the solubility of the microcapsules.

The porosity of powder sample was 0.45% ± 0.28. Both WPI and MCC obey mesoporous materials with pore sizes between 2 and 50 nm, making possible the creation of porous particles. Porous microparticles are widely desired as suitable materials as carriers for the drug delivery system.

FTIR microcapsules analysis showed that no new peaks were formed, which is evidence of physic sorption and weak interactions among the employed encapsulant agents and among grapeseed oil. Additionally, the presence of the characteristic peaks corresponding to the encapsulated fatty acids show that there is a protection provided by the encapsulating materials. Physical adsorption observed between the different materials is provided by secondary interactions such as hydrogen bonds and Van der Waals forces. The foregoing allows us to suppose that the opening of the microcapsules to release the encapsulated grapeseed oil can occur under the conditions of the gastrointestinal system with factors such as pH, temperature, and agitation.

Morphological analysis showed a good thickness of the wall of the microcapsules confirming a correct encapsulation of grapeseed oil. The grapeseed oil (light gray color) was concentrated in the center of the particle, being surrounded by the wall material. The particle size distribution of the microcapsules showed a heterogeneous distribution with about 50% of the microparticles with a size of 10 microns, a peak that represents a predominant size, but only indicated the presence of micrometric particles and exhibited monomodal behavior.

The controlled release result cohered with FTIR results, where physical interactions among wall materials and weak interactions with grapeseed oil were observed. Additionally, the release at pH 1 was consistent with the cellulose reaction at low pH where cellulose tends to bind. The solubility of WPI and MCC was lower than the solubility of WPI, which explains the controlled release of grapeseed oil during the desorption tests.

These microcapsules can be used for a controlled release system based on pH. At low pH values, release is retarded, while high pHs accelerate the release. In this sense, these microcapsules can be used to release bioactive compounds in the small intestine, where pH is above 5.8 and food remains for around 3 h.

## Figures and Tables

**Figure 1 foods-11-03954-f001:**
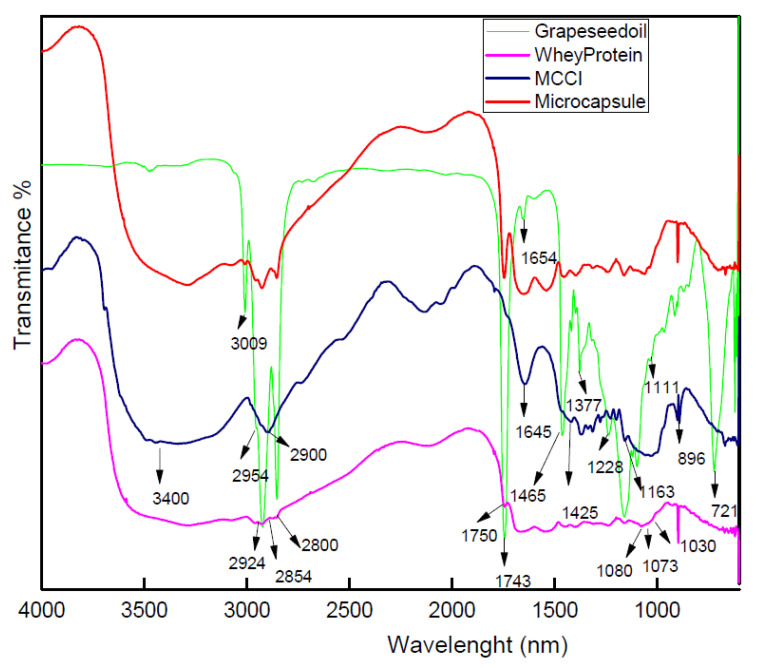
FTIR spectra of samples of WPI, MCC, grapeseed oil, and microcapsules in the wavenumber range of 4000–400 cm^−1^.

**Figure 2 foods-11-03954-f002:**
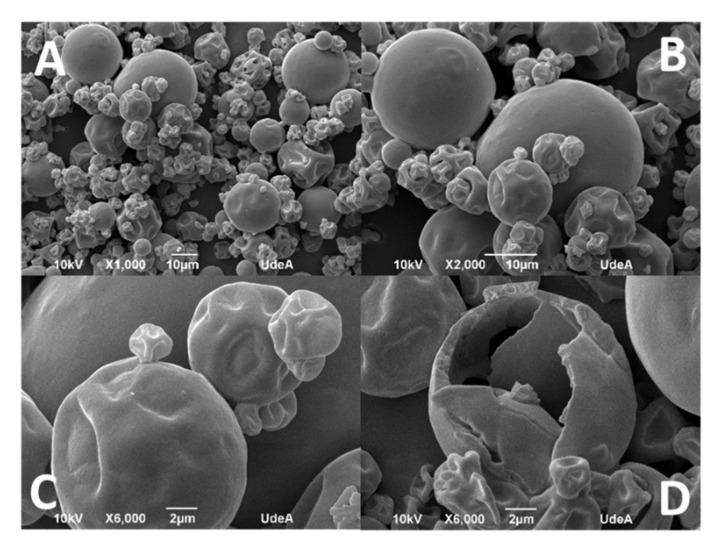
(**A**) X 1000; (**B**) X 2000; (**C**) X 6000; (**D**) the microcapsule’s internal morphology.

**Figure 3 foods-11-03954-f003:**
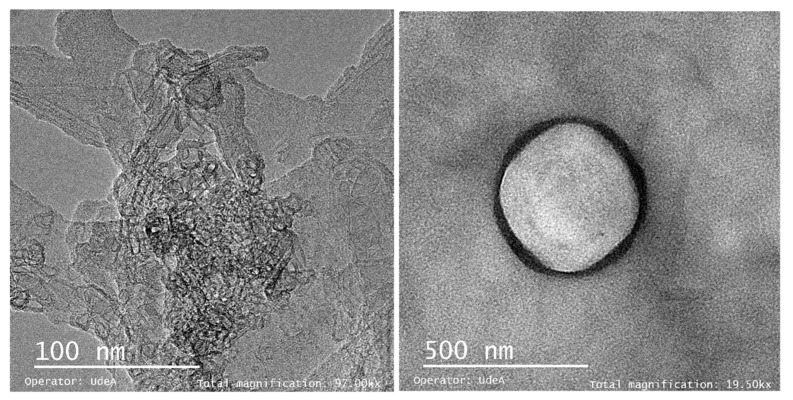
Microcapsule TEM micrograph.

**Figure 4 foods-11-03954-f004:**
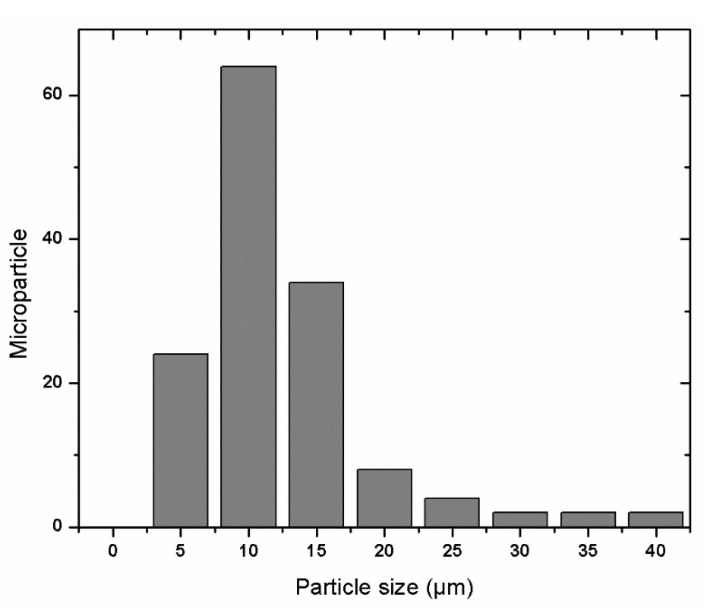
Microcapsule’s particle size distribution.

**Figure 5 foods-11-03954-f005:**
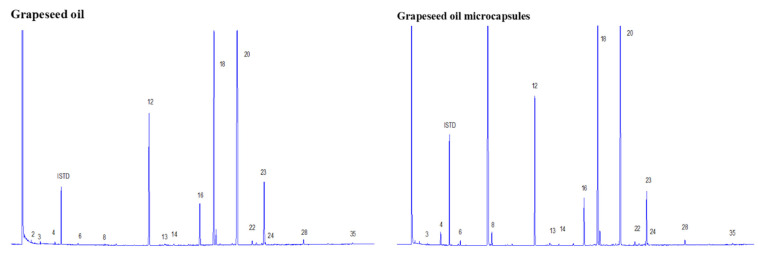
Fatty acid gas chromatographic profiles; left: grapeseed oil; right: grapeseed oil microcapsules.

**Figure 6 foods-11-03954-f006:**
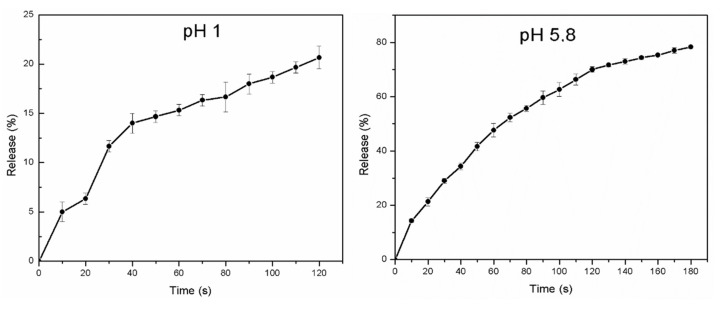
Microcapsule’s controlled release at pH 1 (**left**) and pH 5.8 (**right**).

**Table 1 foods-11-03954-t001:** FTIR functional groups.

Micro Cristalin Celullose MCC	**Setting**	**Waveleght** **(cm^−1^)**	**Vibration of Functional Groups**
	1	3400 to 3500	Stretching of –OH
	6	2900	Groups and CH_2_
	11	1645	Absorption of water
	13	1425	Intermolecular hydrogen of aromatic ring group
	17	1163	C–O–C Stretching
	22	896	C–H Rock vibration
Grapeseed oil	2	3009	C-H stretching vibration of the cis-double bond (=CH)
	4	2954	Asymmetric stretching vibration of methyl (-CH3) group
	5	2924 and 2852	Methylene (-CH_2_) band vibration (symmetric and asymmetric stretching)
	9	1743	Carbonyl (C=O) functional group
	10	1654	Cis C=C
	12	1465	Bending vibration of CH_3_ and CH_2_ aliphatic groups
	14	1417	Rocking vibration of CH bond of cis-disubstituted alkenes
	15	1377	Symmetric bending vibration of CH_3_
	16	1228 and 1155	Vibration of stretching mode from C-O group in esters
	18	1111	vibration of fatty acids (-CH bending and –CH deformation)
	23	721	Methylene (-CH_2_) rocking vibration (overlapping), vibration of cis- disubstituted olefins
Whey Protein	3	Between 3000 and 2800	C-H stretching vibrations of carbonyl groups of triglycerides
	5	2924	stretch vibrations of -C-H(CH_2_) group of fatty acids
	7	2854	stretch vibrations of -C-H(CH_3_) group of fatty acids
	8	1750	C=O stretch of ester or carboxylic acid group
	19	1080	OH stretch of carbohydrates
	20	1073	ν OH
	21	1030	ν C-O of alcohols

**Table 2 foods-11-03954-t002:** Fatty acid profiles of grapeseed oil, encapsulated and non-encapsulated.

N° Peak	Fatty Acid	Grapeseed Oil (%)	Microcapsules-Grapeseed Oil (%)
2	C6:0	0.10	0.00
3	C8:0	0.10	0.08
4	C10:0	0.12	1.11
6	C12:0	0.11	0.30
8	C14:0	0.08	1.04
12	C16:0	11.11	13.41
14	C17:0	0.09	0.14
16	C18:0	3.77	4.58
22	C20:0	0.37	0.37
28	C22:0	0.46	0.51
35	C24:0	0.19	0.09
Total saturated fat		16.509	21.628
13	C16:1	0.09	0.19
18	C18:1n9c	25.37	25.08
24	C20:1n9	0.43	0.21
Total monoinsaturated fat		25.678	25.473
20	C18:2n6c	51.64	47.30
23	27C18:3n3	6.17	5.60
Total polyinsaturated fat		57.813	52.899
Total fat		100	100

## Data Availability

The data presented in this study are available on request from the corresponding author.

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
