# Peer review of "Protection of Mono and Polyunsaturated Fatty Acids from Grapeseed Oil by Spray Drying Using Green Biopolymers as Wall Material"

_foods, 2022, doi:10.3390/foods11243954_

Round 1
Reviewer 1 Report
Elaborate more about MCC in the introduction part.
What is green biopolymer, elaborate in bit detail in introduction part.
On what basis the selection of WPC and MCC is done (3:1) for feed preparation.
Mention about preparation of MCC in part 2.1 of materials and methods.
Write the reference in one format the manuscript, at some point it is in squared brackets while at some point it is written normally without any parenthesis (2.4 Jadafi (2019); 2.5 Jadafi 2019; and [19]; Tan et al. 2005). Please rectify it.
Write full form of FAME in 2.10.
Provide a refence study for encapsulation efficiency in results and discussion part 3.3.
In figure 1 FTIR spectra there are some peaks are missing which are provided in table 1 like peak 1, 3 and 20.
For particle size distribution, author write about SEM in Materials and methods and discuss about TEM in result and discussion part for the same, rectify it.
Less supportive reference studies were presented in support of the entire manuscript. Add some recent supportive bibliography.
Check the style of refence citation in manuscript reference section, it varied a lot.
Check for the grammatical error in the entire manuscript.
Reviewer 2 Report
The following changes are suggested before considering a possible application in this journal:
- The initial abstract must be reduced by putting more emphasis on the innovation points of the proposed work;
- Line 69: "There is a wide variety of encapsulating agents". Authors should describe in detail the most widely used encapsulating agents, highlighting any advantages and problems arising from their use.
- Line 106: "The previous mixture was left to stand between 4 and 12 hours at room temperature". What is the reason for such a wide interval? What does it depend on?
- Please insert list of abbreviations used in this work
- Line 267: please correct "FTIT" with FTIR
- Figure 1: please replace the numbers with the wavelengths of the frequencies more characteristic
- Line 300: please correct Tabla with Table
- The authors investigated the release of encapsulation in relation to the pH of the reaction medium. Have the authors studied the effect of other parameters, e.g. ionic force of the medium, temperature?
Reviewer 3 Report
The manuscript death with the encapsulation of MUFA and PUFA from grape seed oil using biopolymers as wall materials by spray drying method. The manuscript is written poorly with badly constructed sentences. Besides, the methodology section is confusing.
Comments:
1. Revise the English language and grammar of the manuscript.
2. Title: the title does not reflect the content of the manuscript. The authors reported that they have encapsulated MUFA and PUFA from grapeseed soil. However, I have not found any methodology for separating MUFA and PUFA from grape seed oil.
3. The used microcrystalline cellulose (MCC) obtained from green biopolymers as a wall material for encapsulation. However, I have not found in the manuscript the type of biopolymer used for MCC isolation, isolation technique, and also its characterization.
4. Page 3, line 101: section 2.1 materials?. It is confusing. It seems like preparation techniques of MCC from cotton linter sheet. It suggests to change as: Sample preparation/MCC and protein isolation/…
Although this section seems to be the isolation of MCC, the way it has been written is confusing. It suggests to rewrite as: pulping, bleaching, acid hydrolysis and then screening to obtain MCC.
5. Page 3, section 2.3 Spray-drying conditions: change to encapsulation process and elaborate the method of encapsulation.
6. It suggests to re-arrange the methodology section. Suggestion:
3.1 Sample preparation
3.1.1 Isolation of MCC
3.1.2 Whey protein isolation
3.1.3 preparation of the emulsion
3.2 Fatty acids compositions in grape seeds oil
3.2 Encapsulation using the spray-drying method
3.3 Microencapsulation efficiency
3.4 Characterization (list all the characterizations in this section).
3.5 Controlled release
3.6 Statistical analyses
7. Page 5, lines 227-229: Delete these lines.
8. Page 6 line 267: Please write in detail the figure 1 legend. A figure legend should be stand-alone to explain the figure clearly and thoroughly.
8. The discussion of each major finding must be improved.
9. Re-write the consultations quantitatively with the major findings of each experiment.
Round 2
Reviewer 2 Report
The authors have responded exhaustively to the reviewers' comments, therefore the publication is recommended.
Reviewer 3 Report
I am satisfied with the amended manuscript. Therefore, I am recommending to accept the manuscript for publication in the present form of the manuscript.